# Habitat isolation interacts with top-down and bottom-up processes in a seagrass ecosystem

**Elizabeth W. Carroll**[1]*, **Amy L. Freestone**[2]

**1** Department of Biology, Holy Family University, Philadelphia, Pennsylvania, United States of America,
**2** Department of Biology, Temple University, Philadelphia, Pennsylvania, United States of America

\* ecarroll2@holyfamily.edu

**Data Availability Statement:** The data will be found in Dryad using the doi https://doi.org/10.5061/dryad.3xsj3txm3.

**Funding:** This research was supported by a Temple University Fellowship to EC and the National

## Abstract

Habitat loss is accelerating at unprecedented rates, leading to the emergence of smaller, more isolated habitat remnants. Habitat isolation adversely affects many ecological processes independently, but little is known about how habitat isolation may interact with ecosystem processes such as top-down (consumer-driven) and bottom-up (resource-driven) effects. To investigate the interactive influence of habitat isolation, resource availability and consumer distribution and impact on community structure, we tested two hypotheses using invertebrate and algal epibionts on temperate seagrasses, an ecosystem of ecological and conservation importance. First, we hypothesized that habitat isolation will change the structure of the seagrass epibiont community, and isolated patches of seagrass will have lower epibiont biomass and different epibiont community composition than contiguous meadows. Second, we hypothesized that habitat isolation would mediate top-down (i.e., herbivory) and bottom-up (i.e., nutrient enrichment) control for algal epibionts. We used observational studies in natural seagrass patches and experimental artificial seagrass to examine three levels of habitat isolation. We further manipulated top-down and bottom-up processes in artificial seagrass through consumer reductions and nutrient additions, respectively. We indeed found that habitat isolation of seagrass patches decreased epibiont biomass and modified epibiont community composition. This pattern was largely due to dispersal limitation of invertebrate epibionts that resulted in a decline in their abundance and richness in isolated patches. Further, habitat isolation reduced consumer abundances, weakening top-down control of algal epibionts in isolated seagrass patches. Nutrient additions, however, reversed this pattern, and allowed a top-down effect on algal richness to emerge in isolated habitats, demonstrating a complex interaction between patch isolation and top-down and bottom-up processes. Habitat isolation may therefore shape the relative importance of central processes in ecosystems, leading to changes in community composition and food web structure in marine habitats.

## Introduction

Habitat isolation can decrease connectivity and limit dispersal resulting in dramatic effects on natural communities [1–4]. Despite being widely studied in terrestrial landscape ecology,

Science Foundation Division of Ocean Sciences Grants 1225583 and 1434528 to AF. The funders had no role in study design, data collection and analysis, decision to publish, or preparation of the manuscript.

**Competing interests:** The authors have declared that no competing interests exist.

habitat isolation has been described as the most overlooked application of landscape ecology to coastal environments [5, 6]. In marine systems, the biological consequences of habitat isolation on seascapes are not as well understood, especially regarding how habitat isolation may interact with ecological processes. For instance, dispersal limitation can be an important factor in marine systems [7–9]. The larval stage of many marine species, particularly invertebrates, is often short in length and characterized by a small larval size and poor swimming ability, constraining dispersal potential [8, 9]. Understanding habitat isolation and dispersal limitation in marine systems can clarify how continued isolation of habitats influences ocean sustainability [6].

In addition to dispersal, species colonization of habitat is also shaped by the distribution of both consumers and resources. A reduction in the abundance of higher trophic levels can increase abundances of lower trophic levels through weakened top-down control [10]. By contrast, when lower trophic levels, such as primary producers, are constrained by low nutrient availability [i.e., bottom-up control; 11], the abundance of organisms at higher trophic levels will be indirectly reduced due to a limited food supply [12, 13]. The relative importance of top-down and bottom-up effects can be mediated by the diet breadth of consumers [14], community structure [15], as well as the physical environment [16].

Habitat isolation may mediate top-down and bottom-up controls of species abundances [17–19]. Isolation can decrease the ratio of predators to prey, which can weaken the ability of herbivores to control primary producer abundances [20]. Declines in consumer abundances due to isolation have been shown to weaken top-down impacts in aquatic metacommunities [21] and forest fragments [22]. If herbivores are dispersal limited and therefore low in abundance in isolated habitats, primary producers may be released from top-down control and increase in biomass. Alternatively, primary producers may be constrained by limited nutrient availability in isolated habitats thus constraining higher trophic levels as well. Habitat isolation may influence bottom-up control of ecosystems through reduced nutrient retention [23], lower decomposition rates, and nitrogen immobilization [24]. Despite extensive habitat loss in both terrestrial and aquatic ecosystems leading to increased habitat isolation across the globe [25, 26], we have only a rudimentary understanding of how top-down and bottom-up processes may differentially affect communities in isolated versus continuous habitats [27].

Shallow marine habitats dominated by foundation species have been declining at historic rates [28]. Mangrove losses range from 0.16% to as high as 8.08% in some regions per year [29], while coral reef cover has declined by 15% in the past few decades [30]. Kelp forests have declined in 38% of the world's ecoregions [31]. Seagrass habitats are experiencing global loss and are becoming increasingly isolated, disappearing at a rate of 110km$^2$/year since 1980 with global losses nearing 20% since 1880 [25, 28, 32]. This rate of loss is equivalent to declines in mangroves, coral reefs, and rain forests [25], placing seagrass among the most threatened ecosystems. Like many marine foundation species, seagrasses support high species diversity by providing refuge and nursery habitat for invertebrates and fish [33]. The value of seagrass nutrient cycling alone has been estimated at $1.9 trillion annually [25]. Many organisms rely on seagrass epibiont communities (algae and invertebrates that grow directly on seagrass blades) as a source of food [34–36].

Due to limited dispersal distances and high self-retention, small marine invertebrates may be particularly susceptible to habitat isolation [37]. Isolation may therefore be an important but understudied factor that could shape both epibiont invertebrate communities as well as herbivorous mobile consumers such as isopods and amphipods [3, 38] that are known to graze heavily on algal epibionts [35]. Consumer-resource interactions can control algal overgrowth that might otherwise damage the seagrass through photosynthesis inhibition [39, 40], even when algae are not resource limited [41, 42].

How habitat isolation and its interaction with top-down and bottom-up processes influence seagrass communities remains an open, and increasingly urgent question. Our aim was to use seagrass habitat, a system of high conservation importance, to explore two questions: how habitat isolation modifies community structure in a marine ecosystem, and how habitat isolation mediates top-down and bottom-up control of primary producers. To address this aim, we used observational and experimental approaches to study the effects of habitat isolation for seagrass epibiont communities, including both invertebrates and algae, as well as top-down (i.e., herbivory) and bottom-up (i.e., nutrient enrichment) control of algae. First, we hypothesized that habitat isolation will change the structure of the seagrass epibiont community. Specifically, we hypothesized that isolated patches of seagrass will have lower epibiont biomass and different epibiont community composition, likely driven by dispersal limitation of sessile invertebrates. Second, we hypothesized that habitat isolation would mediate top-down (i.e., herbivory) and bottom-up (i.e., nutrient enrichment) control for algal epibionts, an important primary producer in this system. While we expected nutrient enrichment to promote algal epibionts in all habitats, we hypothesized that the invertebrate herbivores of algae would have reduced abundances in isolated habitats, and therefore top-down control would weaken with increasing isolation.

## Methods

In the Barnegat Bay-Little Egg Harbor Estuary, New Jersey, 50–88% of seagrass biomass has been lost since the 1980s [43, 44], making the area well suited to examine the impacts of habitat isolation on seagrass ecosystems. In Island Beach State Park, a preserved area of Barnegat Bay, New Jersey, we selected a large (0.32km$^2$) continuous meadow of *Zostera marina*, the dominant seagrass species in the outer portions of the estuary, surrounded by both seagrass patches and unvegetated sediment for the study site (S1 Fig; 39˚N 47’ 24.8064", 74˚W 6’ 28.476"). We conducted three field studies at this location to test if habitat isolation shapes epibiont community biomass and structure and mediates top-down and bottom-up processes in a seagrass ecosystem: 1) Habitat isolation in natural seagrass; 2) Habitat isolation and consumer-resource interactions and 3) Recruitment. Depth in the study area during our study period (July-August 2013) ranged from 0.5 to 0.9 m, temperature from 21 to 25.3˚C, and salinity from 20 to 25 PSU. Vertical clarity, measured with a Secchi disk, usually reached the sediment, and horizontal clarity ranged from 0.5 to 2.1 m.

### Habitat isolation in natural seagrass

To test the hypothesis that habitat isolation changes seagrass epibiont community structure, we surveyed epibiont communities in the large seagrass meadow and adjacent patches that varied in their level of isolation. We used transects to assess the frequency and distribution of natural seagrass patches around the meadow habitat. In August 2013, during the period of peak productivity in this seasonal seagrass system, we established five 50 meter transects extending from randomly selected starting points 10m within the continuous seagrass meadow into the surrounding matrix of seagrass patches in the study area. At the start of each transect we collected 10 blades from a 0.3m x 0.3m area in the continuous meadow. We then collected 10 blades from each natural seagrass patch greater than 0.3m x 0.3m in area observed along the transects. For analysis, we categorized transect samples into three habitat types: continuous meadow (n = 5), low isolation (0-20m away from the meadow; n = 5) and high isolation (>greater than 20m away from the meadow; n = 7), for a total of 17 samples. In the laboratory, epibionts were identified to the lowest possible taxonomic level (usually species) using a dissecting microscope to characterize epibiont community structure, and abundances of each

group were recorded as the numbers of individuals observed across all ten blades per sample. The 10 clipped natural seagrass blades in each sample (170 blades in total; N = 17) were scraped of epibionts using a razor blade, and epibionts and seagrass were dried separately at 60˚C for three days and weighed for dry biomass. Epibionts and seagrass were then incinerated at 500˚C for four hours and reweighed to determine ash free dry weight (AFDW). We tested the effect of habitat type (3 levels) on the biomass of epibionts using ANOVA (JMP® v.16 2014). Epibiont community composition was analyzed using a permutational multivariate ANOVA [PERMANOVA, 45] based on a Bray-Curtis similarity matrix with the fixed effect of habitat type. Initial models included transect as a random effect, but transect was not significant (P>0.8) and was removed from the final models.

## Habitat isolation and consumer-resource experiment

To test the hypotheses that habitat isolation mediates top-down and bottom-up processes, we ran a field experiment in July 2013. We used artificial seagrass units (ASUs) as our experimental unit to avoid potentially confounding factors such as seagrass density and blade length. ASUs have been extensively used as a standardized mimic of natural seagrass and have been shown to attract similar species [46]. ASUs were constructed of green polypropylene ribbon attached to plastic mesh. ASU area (0.6m x 0.6m) and blade length (30 cm) were based on mean measurements of seagrasses and patch dimensions from surveys of patches at the study site. Blade density was determined from the literature (239–466 shoots m$^{-2}$) and was 308 shoots m$^{-2}$ (ASUs contained 111 seagrass blades) [43]. Each blade was 4.7mm wide.

Twenty ASUs were placed at each of three isolation levels relative to the continuous meadow (N = 60; Fig 1): 1) 10m within the meadow, 2) 10m from the edge of the meadow (i.e., low isolation), and 3) 25m from the edge of the meadow (i.e., high isolation). The 10m and 25m isolation treatments represent the 25th and 75th percentiles for the distances of natural patches to the meadow based on surveys of meadow-patch spatial arrangements at the site. Within isolation levels, ASU patches were spaced 2.5m away from each other. We manipulated

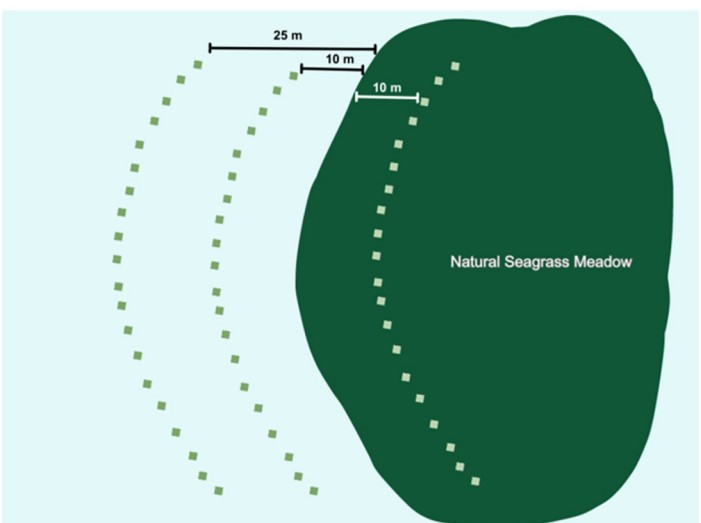

**Fig 1. A diagram of the study design for the ASU experiment shows the approximate locations of ASUs relative to the natural seagrass meadow.**

bottom-up and top-down factors through nutrient additions and herbivore reductions, respectively. The following four treatments were haphazardly applied to five replicate ASUs within each isolation level: 1) herbivore reduction, 2) nutrient addition, 3) a combination of herbivore reduction and nutrient addition, and 4) no manipulation (i.e., ambient conditions). To reduce herbivores, primarily small invertebrate grazers in this system, we incorporated carbaryl (Sevin SL), an insecticide that inhibits arthropods (an abundant consumer in this ecosystem) through neurodisruption, into plaster to form 7.5% carbaryl blocks [47]. Previous studies have demonstrated the efficacy of carbaryl in reducing invertebrate abundances within a 30cm radius [39, 47]. We attached one block to a stake 15cm above the sediment in the center of each ASU. As the plaster block dissolves, carbaryl is released into the water column deterring invertebrate herbivores. Carbaryl blocks were replaced on a weekly basis, prior to complete dissolution. The herbivore reduction treatment was shown to be effective, causing a reduction in the abundance of herbivores by 50% (herbivore reduction: $F_{1/56}$ = 4.96, P = 0.031). To manipulate nutrient levels, we attached 300g of Scott's™ continuous release plant fertilizer (10-10-10; N:P:K) enclosed in a nylon mesh bag to a stake in the center of each ASU, and replaced the fertilizer on a weekly basis (prior to complete dissolution). Adding slow release fertilizer in this manner has been shown to increase local water column nutrient levels in similar experiments [48, 49]. For the combined herbivore reduction and nutrient addition treatment we added both a carbaryl block and a nutrient bag to the stake in the ASU. This experiment was deployed for eight weeks, accounting for a large portion of the seagrass growing season in this region [43].

At the completion of the experiment, we assessed both the herbivore abundance and the epibiont community in each experimental plot. We sampled the ASUs for herbivore abundance using a small dipnet to make three sweeps across the plot and preserved the samples using 70% ethanol for enumeration in the laboratory. To assess the epibiont community, ten blades were then clipped from each ASU within a 30cm radius from the experimental treatment. One replicate was not recovered from the low isolation treatment (N = 59). A subset of three blades from three replicates of each treatment (N = 36) was used to determine epibiont community structure, richness, and abundance. Epibionts were identified to the lowest taxonomic group possible (usually species) using a dissecting microscope, and abundances of each group were recorded as the numbers of individuals observed across all three blades per replicate. All of the 10 clipped ASU blades in each replicate (N = 59) were scraped of epibionts using a razor blade, and epibionts were dried at 60°C for three days and weighed for dry biomass. Epibionts were then incinerated at 500°C for four hours and reweighed to determine epibiont ash free dry weight (AFDW) per sample.

The effects of isolation (meadow, low isolation, high isolation), nutrient addition (addition vs. ambient), and herbivore reduction (reduced vs. ambient) manipulations on the epibiont biomass, richness, and abundance were analyzed using analysis of variance (ANOVA, JMP® v.11 2007) with all possible interactions among terms. Variation in community composition was modeled with the same main effects and interactions using a permutational multivariate ANOVA [PERMANOVA, 45] based on a Bray-Curtis similarity matrix. To identify what species drove differences in community similarity, we used a SIMPER analysis to identify the species that were most important in structuring the observed patterns [PERMANOVA, 45]. We then tested for effects of isolation, nutrient addition, and herbivore reduction manipulations as above on algae richness and abundance and invertebrate richness and abundance using analysis of variance (ANOVA, JMP® v.11 2007) with all possible interactions among terms to elucidate how habitat isolation and consumer-resource dynamics may interact to influence these different subsets of the epibiont community.

### Recruitment

To determine if dispersal limitation of epibionts into isolated patches influenced their abundance on seagrass, we measured epibiont recruitment into experimental patches for eight weeks beginning June 2014, during the peak settlement period for this region. While recruitment measurements integrate both differences in dispersal as well post-settlement mortality due to predation, prior research suggests very low levels of predation on newly-settled recruits in this area. We placed five small ASUs (0.15m x 0.15m) which contained 10 blades at each of the three experimental isolation levels (10m into the meadow, low isolation, and high isolation) for a total of 15 ASUs. Each small recruitment ASU was set inside a larger 0.6m x 0.6m ASU. This set of ASUs was in addition to the 59 experimental ASUs and not included in the previously outlined experiment. Recruitment ASUs were retrieved every two weeks and replaced with new ASUs over an eight-week time frame. The epibiont recruits were identified to the lowest taxonomic group possible (usually species), and abundances of each group were recorded as the numbers of individuals observed across all 10 blades per replicate. We used linear mixed models to test for differences in invertebrate and algae recruit abundance and richness with a fixed factor of habitat isolation (meadow, low isolation, high isolation) and date (4 two-week intervals) as a random effect (JMP® v.11 2007).

## Results

### Effects of isolation on epibiont community

Habitat isolation reduced epibiont biomass and altered community structure in both natural and artificial seagrass. In natural seagrass, epibiont biomass was lower in both low isolation and high isolation patches than meadow patches (Fig 2, ANOVA, $F_{2/14}$ = 6.69, P = 0.0092). Epibiont biomass in low isolation patches was not significantly different from meadow or high isolation patches. Habitat isolation also affected the epibiont community composition in natural seagrass, with low isolation patches being different from the meadow (PERMANOVA Pseudo-$F_{1/15}$: 2.13, P[perm] = 0.0327; pairwise tests: meadow-low isolation P[perm] = 0.049; meadow-high isolation and low-high isolation P[perm]>0.05).

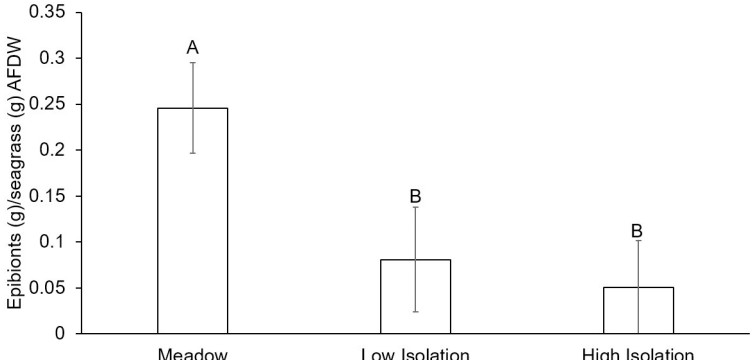

**Fig 2. Epibiont biomass in natural seagrass meadow and isolated patches.** In natural seagrass, low and high isolation seagrass patches had lower epibiont biomass than meadow seagrass. Epibionts are expressed as grams of ash free dry weight per gram of ash free dry weight (AFDW) of seagrass. Meadow are samples from 10m within a continuous meadow, low isolation are patches 0-20m away from the meadow, and high isolation are patches >greater than 20m away from the meadow. Error bars indicate standard error and letters above bars indicate significant differences among treatments at α = 0.05, as determined using a Tukey's HSD Test.

In standardized artificial seagrass, habitat isolation also lowered epibiont biomass and abundance, and altered epibiont community structure (Table 1). Under ambient grazing pressure, ASUs at both low and high isolation had less epibiont biomass (AFDW) than ASUs placed in the meadow (Fig 3, Table 1). Interestingly, herbivore reduction dampened this pattern (Fig 3). Habitat isolation also reduced total epibiont abundances (Table 1) and shaped community composition (Fig 4, PERMANOVA, Pseudo-$F_{2/24}$ = 2.35, P[perm] = 0.0461, pairwise tests: meadow-high isolation, $F_{2/24}$ = 2.03, P[perm] = 0.0243; meadow-low isolation, $F_{2/24}$ = 1.56, P[perm] = 0.0648; low-high isolation, P>0.05). In addition to these isolation effects, nutrient addition reduced total epibiont abundances (Table 1) and altered community composition (Psuedo-$F_{1/24}$ = 4.77, P[Perm] = 0.0088), as did grazing (Pseudo-$F_{1/24}$ = 3.69, P[perm] = 0.017; all interactions P>0.05).

Changes in epibiont community structure and the decline in total epibiont abundances with isolation were driven by a decline in the abundance of invertebrates from meadows to both isolated habitats (Fig 5; Table 1). Declines in invertebrate species such as *Spirorbis spirillum*, *Spirorbis violaceus* (both tube-building worms), and *Botryllus schlosseri* (compound ascidian) with isolation accounted for 83–84% of variation between the meadow and both types of isolated habitats (SIMPER analysis). Invertebrate abundance also declined with nutrient addition (Table 1). There was no significant effect of any factor on epibiont invertebrate richness.

Results from the recruitment experiment support our findings that habitat isolation decreased the abundance and richness of invertebrate epibionts recruiting into seagrass habitat. We found lower invertebrate recruit abundance and richness settling on seagrass in isolated patches (both low and high isolation) than meadow patches across the experimental duration (Fig 6, ANOVA, abundance: $R^2_{adj}$ = 0.86, $F_{11/46}$ = 33.32, P<0.001; richness: $R^2_{adj}$ = 0.42, N = 58, $F_{11/46}$ = 4.71, P = <0.001; date: p>0.05). The abundance and richness of algae recruits were not affected by isolation (P>0.05).

### Effects of isolation on top-down and bottom-up control of algal epibionts

As hypothesized, isolation mediated top-down and bottom-up control of algae in seagrass epibiont communities. First, herbivore abundances were lower in high isolation patches (Fig 7; Table 1), likely weakening top-down influences in isolated habitats. Further, herbivore reductions, isolation, and nutrient additions interacted to shape patterns of algae richness, but not abundance (Fig 8; Table 1), and underlay patterns in overall epibiont richness (Table 1). In meadow patches, herbivores reduced algae richness only under ambient nutrient levels. When nutrients were added, there was no observable effect of herbivores on algae richness. By contrast, in high isolation habitats herbivores reduced algae richness only in combination with nutrient additions. At low isolation we observed a hybrid response, where herbivores reduced algae richness at both ambient and increased nutrient levels. Therefore, at ambient nutrient levels we observed stronger top-down control in meadows and weaker top-down control in high isolation habitats. Bottom-up effects of nutrient additions dampened top-down effects in meadows but allowed them to emerge in isolated habitats.

### Discussion

Small-scale changes to the spatial structure of a seascape can have important ecological consequences for the structure and composition of marine communities. We found that habitat isolation had a clear impact on epibiont biomass, abundance, and community structure. Both artificial and natural seagrass patches isolated from the continuous meadow had over two to four times lower epibiont biomass and abundance, and different community composition than habitat within the meadow. These results suggest that even modest separation of a patch

**Table 1. Results from the full model analysis of the influence of habitation isolation, herbivore reduction, and nutrient additions on epibionts, algae, and invertebrates on artificial seagrass.**

| Artificial Seagrass | Full Model | | | | Isolation | | | Nutrient addition | | | Herbivore reduction | | | Isolation*Nutrient addition | | | Isolation*Herbivore reduction | | | Nutrient addition*Herbivore reduction | | | Isolation*Nutrient addition*Herbivore reduction | | |
|---|---|---|---|---|---|---|---|---|---|---|---|---|---|---|---|---|---|---|---|---|---|---|---|---|---|---|
| | $R^2_{adj}$ | DF | F | P | DF | F | P | DF | F | P | DF | F | P | DF | F | P | DF | F | P | DF | F | P | DF | F | P |
| Epibiont biomass | 0.47 | 11/48 | 3.97 | <0.001 | 2 | 15.1 | <0.001 | | | NS | | | NS | | | NS | 2 | 3.49 | 0.0384 | | | NS | | | NS |
| Epibiont abundance | 0.30 | 11/24 | 2.38 | 0.0364 | 2 | 5.64 | 0.0098 | 1 | 6.94 | 0.0145 | | | NS | | | NS | | | NS | | | NS | | | NS |
| Epibiont richness | 0.36 | 11/24 | 2.80 | 0.0169 | | | NS | | | NS | 1 | 15.4 | <0.001 | 2 | 4.58 | 0.0207 | | | NS | | | NS | | | NS |
| Algae richness | 0.36 | 11/24 | 2.82 | 0.0162 | | | NS | | | NS | 1 | 18.7 | <0.001 | | | NS | | | NS | | | NS | 2 | 3.41 | 0.0497 |
| Algae abundance | NS | | | | | | | | | | | | | | | | | | | | | | | | |
| Invertebrate richness | NS | | | | | | | | | | | | | | | | | | | | | | | | |
| Invertebrate abundance | 0.36 | 11/24 | 2.76 | 0.0181 | 2 | 6.25 | 0.0065 | 1 | 8.14 | 0.0088 | | | NS | | | NS | | | NS | | | NS | | | NS |
| Herbivore abundance | 0.28 | 11/45 | 3.06 | 0.0039 | 2 | 9.74 | <0.001 | | | NS | 1 | 4.96 | 0.0311 | | | NS | | | NS | | | NS | | | NS |

NS indicates non-significant.

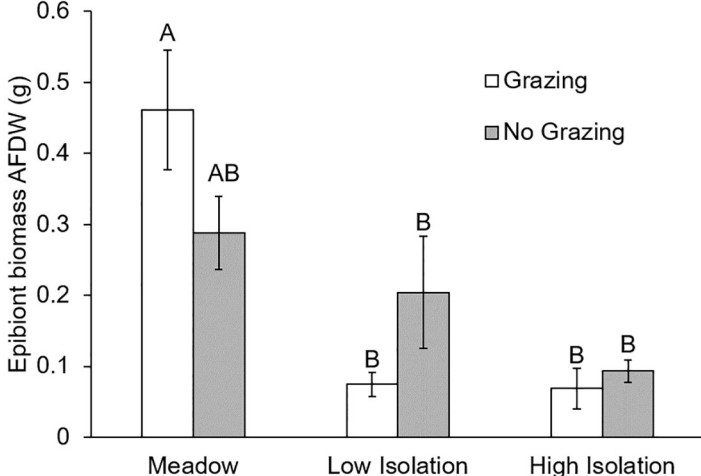

**Fig 3. Epibiont biomass on artificial seagrass in a meadow and isolated patches under ambient and reduced grazing treatments.** Habitat isolation reduced epibiont biomass (which includes both algae and invertebrate epibionts) on artificial seagrass under ambient grazing pressure. Grazer reduction dampened this pattern. Meadow are ASUs 10m within the continuous meadow, low isolation are patches 10m away from the meadow, high isolation are patches 25m away from the meadow. Error bars indicate standard error and different letters above bars indicate significant differences among treatments at α = 0.05, as determined using a Tukey's HSD Test.

from a continuous meadow and thus an epibiont source population can impact the abundance and species that disperse through the matrix. Habitat isolation had particularly strong effects on invertebrate epibionts, with decreases in invertebrate epibionts driving the changes in community composition and abundance. Recruitment patterns support the assertion that dispersal limitation is a likely mechanism for declines in epibiont abundances. The abundance and richness of invertebrates settling in seagrass habitat declined in isolated patches relative to the

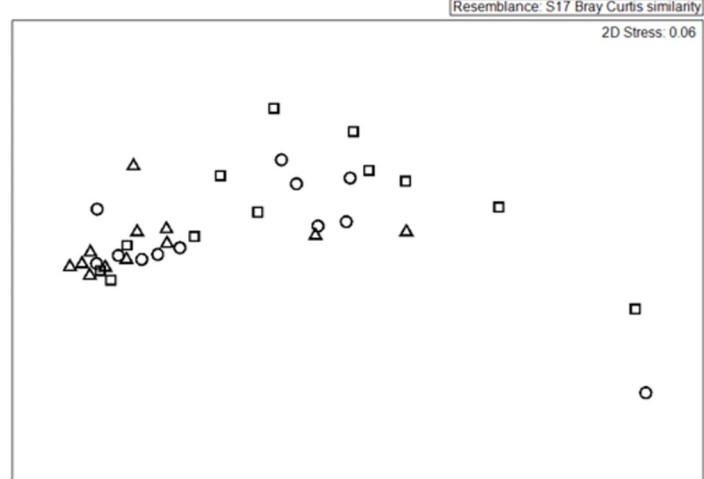

**Fig 4. Non-metric multidimensional scaling ordination of epibiont communities on artificial seagrass in a meadow and isolated patches.** On artificial seagrass total epibiont community structure (including algae and invertebrates) is different in isolated habitats than meadow habitats. The ordination reflects Bray-Curtis similarity measures for epibiont communities in patches 10m within the meadow (triangles), low isolation (10m away from the meadow) (circles), and high isolation (25m away from the meadow) (squares) patches. Points closer together are more similar in composition.

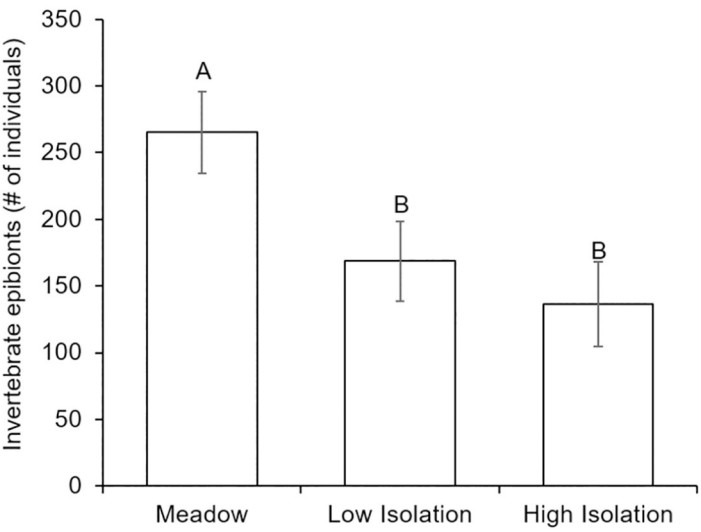

**Fig 5. Abundance of invertebrate epibionts on artificial seagrass in a meadow and isolated patches.** Habitat isolation reduced the number of invertebrate epibiont individuals on artificial seagrass. Meadow are ASUs 10m within the continuous meadow, low isolation are patches 10m away from the meadow, high isolation are patches 25m away from the meadow. Error bars indicate standard error and letters above bars indicate significant differences among treatments at α = 0.05, as determined using a Tukey's HSD test.

meadow. Sessile invertebrates are likely more sensitive to isolation due to their short pelagic stage and weak swimming ability as larvae [8, 9].

Indeed, many species of sessile invertebrate groups such as cnidarians, bryozoans, ascidians, and sponges often settle within 10m of their source [9, 50–53], observations that align with our experimental results. In particular, ascidians tend to have the shortest dispersal potential, as many are limited by their larval yolk sac as a nutrient source, and compound ascidians often settle within minutes [51]. We found declines in *B. schlosseri*, a compound ascidian in the system, accounted for a large portion of the dissimilarity in community structure between meadow and isolated habitats. While many marine organisms such as fish, algae, and mollusks may have dispersal potentials on the order of 100s of meters [8, 9], small isolation distances in marine systems can be a barrier to dispersal for many sessile invertebrates, and these barriers can have important community-level consequences.

Habitat isolation can also have important impacts on top-down dynamics. Herbivores were less abundant in highly isolated patches, and subsequently the extent of habitat isolation determined the influence of herbivory on the primary producers in this system, algal epibionts. Under ambient nutrient conditions, consumers reduced algal epibiont richness by half in meadows, with a significant but slightly more modest effect size in low isolation patches. In contrast, there was no effect of consumers on algal richness in highly isolated patches. Therefore, at ambient resource levels, top-down processes were substantially weakened with increasing habitat isolation likely due to a concomitant decline in herbivore abundances, a result that aligns with theoretical predictions and previous empirical findings in aquatic and terrestrial systems [21, 22].

Further, bottom-up processes were found to mediate the influence of top-down processes on primary producers. While nutrient additions did not increase algal richness in comparison to controls, nutrients did dampen the effect of herbivores within the meadow, while allowing an herbivore effect to emerge in highly isolated patches. Interestingly, low isolation habitats produced an intermediate effect, with grazers reducing algal richness in both ambient and

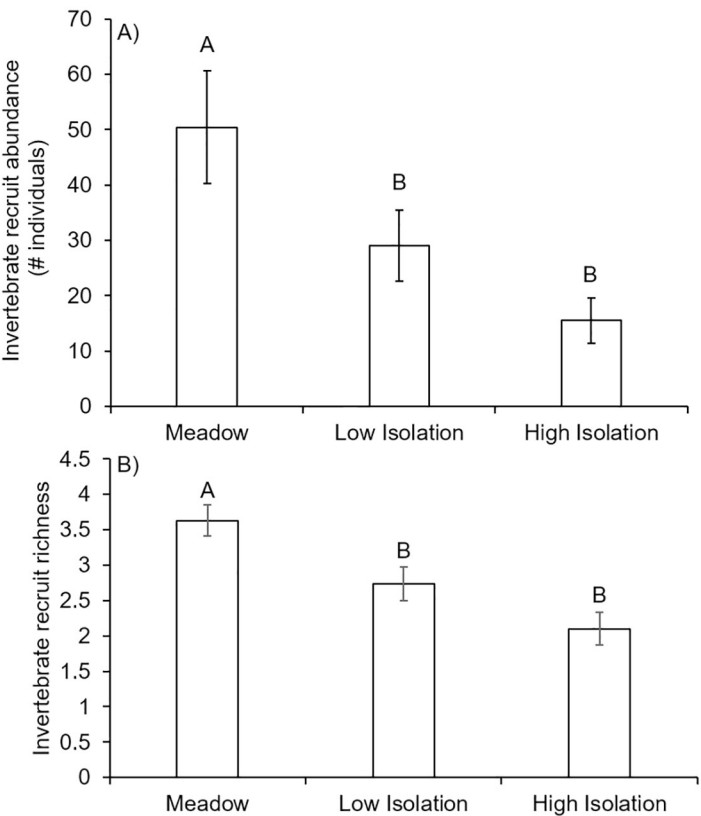

**Fig 6. Recruitment of invertebrate epibiont (A) abundances and (B) richness on artificial seagrass in a meadow and isolated patches.** Both abundance and richness were higher in meadow habitats than isolated habitats. Abundances are expressed as the mean number of individuals recruiting to each habitat type per two-week period across the 8-week experimental timeframe. Richness is expressed as the mean number of species recruiting to each habitat type per two-week period across the 8-week experimental timeframe. Meadow are ASUs 10m within the continuous meadow, low isolation are patches 10m away from the meadow, high isolation are patches 25m away from the meadow. Error bars indicate standard error (N = 58) and letters above bars indicate significant differences among treatments at α = 0.05.

nutrient addition treatments. In meadows, nutrient additions may have increased the occurrence of unpalatable taxa or increased palatable taxa above a point at which grazers could have an observable effect [i.e., consumer swamping, 54]. In isolated habitats, an increase in the occurrence of palatable taxa may have allowed a consumer effect to emerge even though herbivore abundances there were low. Indeed, nutrient limitation can occur in seagrass beds and alter algal epibiont composition and abundance [55, 56], and we show that these changes may also mediate top-down effects.

We conducted this factorial experiment using artificial seagrass, and while differences in epibiont recruitment could occur between artificial and natural seagrass, studies have shown that both substrates can support similar species pools and ecological dynamics, including top-down consumer effects on the epibiont community [57]. Effect sizes, however, can differ between these substrates depending on the metric being examined, and therefore while the directionality of these patterns is likely to translate to natural seagrass, effect sizes may differ. Nonetheless, our results highlight the complex interplay between top-down and bottom-up processes and habitat isolation that structure epibiont composition in this system.

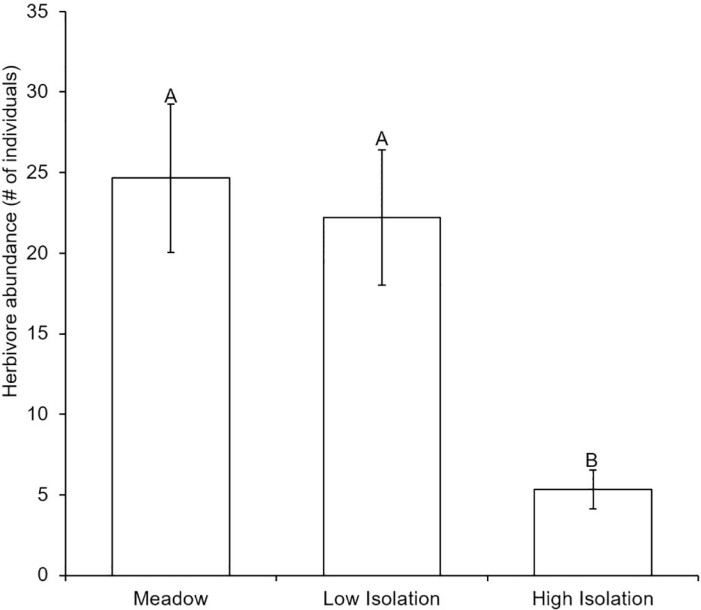

**Fig 7. Herbivore abundance on artificial seagrass in a meadow and isolated patches.** Herbivore abundance declined in isolated habitats. Mean number of consumers sampled from ASUs in meadow and isolated habitats are shown. Meadow are ASUs 10m within the continuous meadow, low isolation are patches 10m away from the meadow, high isolation are patches 25m away from the meadow. Error bars indicate standard error and letters above bars indicate significant differences among treatments at α = 0.05, as determined using a Tukey's HSD test.

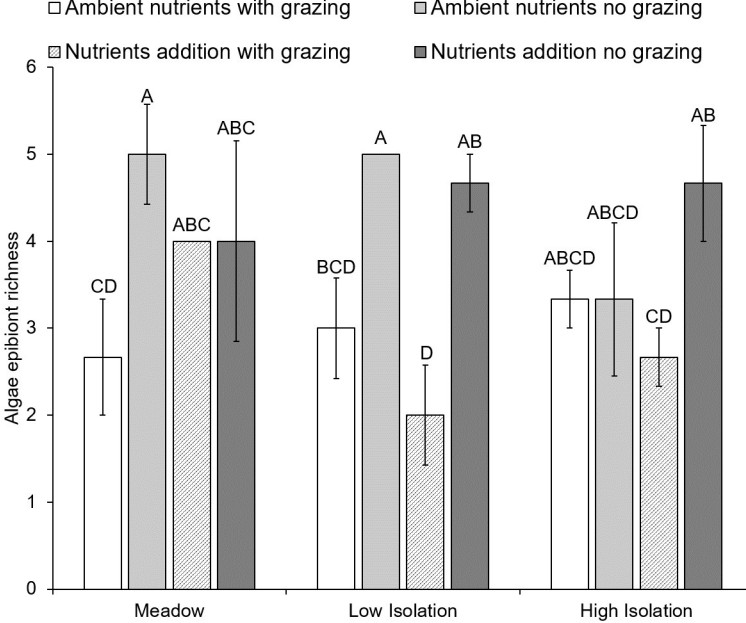

**Fig 8. Algal epibiont richness on artificial seagrass in a meadow and isolated patches under grazing and nutrient treatments.** The effect of herbivore (i.e., grazer) reduction and nutrient additions on algal richness was mediated by habitat isolation. Meadow are ASUs 10m within the continuous meadow, low isolation are patches 10m away from the meadow, high isolation are patches 25m away from the meadow. Error bars indicate standard error and letters above pairs indicate significant differences among treatments at α = 0.05, as determined using a Tukey's HSD test.

Overall, interactions between habitat isolation and top-down and bottom-up processes can produce very different epibiont communities on seagrass, which may alter both primary and secondary production. Epibionts are an important basal food resource in seagrass ecosystems [34, 36], and the abundance of some predators such as fish and mud crabs can decrease in response to low epibiont biomass [58]. Changes in the abundance and composition of epibiont communities may therefore influence valuable ecosystem services such as secondary production in seagrass beds [59]. Our results emphasize the need for a greater understanding of how fundamental ecological processes may vary spatially in marine systems, especially as marine habitat loss continues to accelerate.

## Supporting information

**S1 Fig. Map of study site.** The study location (39˚N 47' 24.8064", 74˚W 6' 28.476") within Island Beach State Park, New Jersey, USA is indicated with a black pin.
(TIF)

## Acknowledgments

We thank K. Bezik, C. Carnivale, R. Czaja, C. Hines, R. Kwait, J. Linnell, D. Lopez, P. Maleszewski, and B. Smith for laboratory and field assistance, A. Strange for assistance creating the study site map, Island Beach State Park, NJ for accommodating our research, and E. Cordes, M. Russell, and R. Sanders for useful feedback during the development of the project and manuscript.

## Author Contributions

**Conceptualization:** Elizabeth W. Carroll, Amy L. Freestone.

**Data curation:** Elizabeth W. Carroll.

**Formal analysis:** Elizabeth W. Carroll.

**Funding acquisition:** Amy L. Freestone.

**Writing – original draft:** Elizabeth W. Carroll, Amy L. Freestone.

**Writing – review & editing:** Elizabeth W. Carroll, Amy L. Freestone.

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
