## [Decision Letter · Decision Letter 0]

17 Apr 2023

PONE-D-23-00408Habitat isolation interacts with top-down and bottom-up processes in a seagrass ecosystemPLOS ONE

Dear Dr. Carroll,

Thank you for submitting your manuscript to PLOS ONE. After careful consideration, we feel that it has merit but does not fully meet PLOS ONE’s publication criteria as it currently stands. Therefore, we invite you to submit a revised version of the manuscript that addresses the points raised during the review process.

Both reviewers highlighted methodological issues (sampling design, use of transects, etc.) that must be fixed before publication. In addition, I noted that “ppt” has not been in use for salinity measures since decades: please check if you can adopt PSU, instead.

We look forward to receiving your revised manuscript.

Kind regards,

Carlo Nike Bianchi

Academic Editor

PLOS ONE

Journal Requirements:

Reviewers' comments:

Reviewer's Responses to Questions

**Comments to the Author**

1. Is the manuscript technically sound, and do the data support the conclusions?

Reviewer #1: Yes

Reviewer #2: Yes

2. Has the statistical analysis been performed appropriately and rigorously? 

Reviewer #1: Yes

Reviewer #2: Yes

3. Have the authors made all data underlying the findings in their manuscript fully available?

Reviewer #1: Yes

Reviewer #2: Yes

4. Is the manuscript presented in an intelligible fashion and written in standard English?

Reviewer #1: Yes

Reviewer #2: Yes

5. Review Comments to the Author

Reviewer #1: The manuscript “Habitat isolation interacts with top-down and bottom-up processes in a seagrass ecosystem” showcase an interesting and current topic worthy to be object of detailed studies. Hypotheses are well argued in the introduction; however, I believe the authors could provide a clearer description of the sampling design used and add a cautionary paragraph in the discussion to critically consider weakness and results of the study.

Below you will find more detailed comments and suggestions.

Lines 77-81: I suggest to add more recent and updated references to support these statements.

Lines 113-123: It is not clear where are located the three field sites in which the study has been carried out. Are all the three sites included in the Island Beach State Park meadow? Maybe a map and a more detailed description of the study area could help.

Lines 127-129: were the five transects repeated in each field sites (x 3)? The sampling has been carried out in August, are epibionts affected by the seasonality?

Lines 129-132: The sampling design is not clear, 10 blades at the beginning of each transect so totally 50 blades? X 3 site (150)? They represent the sampling effort for the “continuous meadow”, and what about the “isolated” and “very isolated”? 18 patches have been found, totally? How many isolated and very isolated? Please clarify the sampling design and the sampling effort.

Lines 148-151: If it is available, I suggest to add a picture of the placed ASUs.

Lines: 153-154: How many ASUs at each isolation level?

Lines 234-235: I suggest to not speak of the results in the caption of the figure (applies to all figure captions).

Lines 256-259: same comment (see above). As you always consider three levels of isolation, I suggest to uniform them in all the graphs (i.e., meadow, low isolation, high isolation), then you can specify in the caption to which distance they refer to.

Lines 320-321: add references.

Reviewer #2: The paper by Carroll and Freestone aim at evaluating how habitat isolation modifies community structure and mediates top-down and bottom-up control of primary producers in a marine seagrass system. The topic is interesting and the study is rightly conducted, as data are to be published, although some changes are needed to improve the paper. A list of comment/suggestions is reported below.

Line 95: “Our aim was to use seagrass meadows, a system of high conservation importance, to explore two questions”

Line 121: “temperature from 21 to 25.3°C, and salinity from 20 to 25 ppt” temporal reference for these values should be given

Lines 124-142, Habitat isolation in natural seagrass: the sampling method should be better clarified.

How many replicates for each condition? Five? What is the logic to use transects if these latter are not considered in the analyses? Why do not consider three habitats and collect five replicates for each habitats? Transects should be justified or removed

Line 143, Habitat isolation and consumer-resource experiment: the use of artificial blades could potentially modify the recruitment pattern compared to natural blades. This aspect should be at least considered in the discussion.

Lines 171: The analyses of nutrient levels in water column in ASU and control plots are needed to evaluate the nutrient increase; if the authors have no data, they should at least provide bibliographic references about the use of the same method to quantify the nutrient enhancing in seagrass systems (e.g Journal of Sea Research 63 (2010) 173-179; Scientific Reports (2017) 7, 13732)

6. PLOS authors have the option to publish the peer review history of their article (what does this mean?). If published, this will include your full peer review and any attached files.

Reviewer #1: No

Reviewer #2: No

---

## [Author Response · Author response to Decision Letter 0]

31 May 2023

May 31, 2023

Dear Dr. Bianchi,

 Thank you very much for the opportunity to submit a revised manuscript for publication in PLOS ONE. We appreciate the comments we received during the review process, and we have aimed to address these points in our revised submission. Please see below for our point-by-point responses to reviewer comments in italics. We feel these revisions improved the clarity of the manuscript. 

 In addition, we wanted to clarify that all data used in this manuscript are publicly available in the Dryad data repository at the following DOI: 10.5061/dryad.3xsj3txm3. We appreciate you updating our Data Availability statement accordingly.

Thank you again for your consideration of our manuscript for publication in PLOS ONE.

Sincerely,

Elizabeth Carroll

Dear Dr. Carroll,

Thank you for submitting your manuscript to PLOS ONE. After careful consideration, we feel that it has merit but does not fully meet PLOS ONE’s publication criteria as it currently stands. Therefore, we invite you to submit a revised version of the manuscript that addresses the points raised during the review process.

Both reviewers highlighted methodological issues (sampling design, use of transects, etc.) that must be fixed before publication. In addition, I noted that “ppt” has not been in use for salinity measures since decades: please check if you can adopt PSU, instead.

We have adopted the use of PSU in the text. Given the temperature range for our site, no conversion was necessary.

We look forward to receiving your revised manuscript.

Kind regards,

Carlo Nike Bianchi

Academic Editor

PLOS ONE

Journal Requirements:

 Our manuscript now adheres to these requirements.

 No human subjects were used in this research.

All data used in this research are publicly available, and we have included this information in our cover letter above.

Reviewers' comments:

Reviewer's Responses to Questions

Comments to the Author

1. Is the manuscript technically sound, and do the data support the conclusions?

Reviewer #1: Yes

Reviewer #2: Yes

2. Has the statistical analysis been performed appropriately and rigorously?

Reviewer #1: Yes

Reviewer #2: Yes

3. Have the authors made all data underlying the findings in their manuscript fully available?

Reviewer #1: Yes

Reviewer #2: Yes

4. Is the manuscript presented in an intelligible fashion and written in standard English?

Reviewer #1: Yes

Reviewer #2: Yes

5. Review Comments to the Author

Reviewer #1: The manuscript “Habitat isolation interacts with top-down and bottom-up processes in a seagrass ecosystem” showcase an interesting and current topic worthy to be object of detailed studies. Hypotheses are well argued in the introduction; however, I believe the authors could provide a clearer description of the sampling design used and add a cautionary paragraph in the discussion to critically consider weakness and results of the study.

Below you will find more detailed comments and suggestions.

 Thank you for these comments. Please see our responses below.

Lines 77-81: I suggest to add more recent and updated references to support these statements.

Thank you for your suggestion. We’ve updated the manuscript to include additional recent papers which quantify global losses of key marine foundation species such as kelp, coral, mangroves, and seagrass.

Lines 113-123: It is not clear where are located the three field sites in which the study has been carried out. Are all the three sites included in the Island Beach State Park meadow? Maybe a map and a more detailed description of the study area could help.

Thank you for identifying this point of confusion. We completed three field studies at the same field site in Island Beach State Park. We updated the manuscript to include more detailed information about this study location, and we now include a map of our study site as a supplemental figure.

Lines 127-129: were the five transects repeated in each field sites (x 3)? The sampling has been carried out in August, are epibionts affected by the seasonality?

There is only one field site which consists of a large continuous meadow of seagrass surrounded by patches of seagrass. Transects ran from the meadow into the patches. We have revised the text in this section to improve clarity. Epibiont recruitment can be influenced by seasonality. Recruitment occurs from May to October in this region, with peak recruitment occurring during the summer months (June-August). Therefore, the epibiont composition in August reflects the period of peak recruitment and productivity. We modified the manuscript text in this section to clarify this point.

Lines 129-132: The sampling design is not clear, 10 blades at the beginning of each transect so totally 50 blades? X 3 site (150)? They represent the sampling effort for the “continuous meadow”, and what about the “isolated” and “very isolated”? 18 patches have been found, totally? How many isolated and very isolated? Please clarify the sampling design and the sampling effort.

We’ve revised this section to provide more detail on the sampling effort for the natural seagrass analysis.

Lines 148-151: If it is available, I suggest to add a picture of the placed ASUs.

Thank you for this suggestion, but unfortunately, we do not have a publication-quality image to accompany our paper. However, ASUs are a commonly used approach in seagrass studies, and there are other images available online that demonstrate this approach.

Lines: 153-154: How many ASUs at each isolation level?

We’ve clarified in the manuscript that twenty ASUs were placed at each of three isolation levels for a total of 60 ASUs. 

We have also included S2 Fig as a supplementary file that shows the spatial arrangement of ASUs in our study.

Lines 234-235: I suggest to not speak of the results in the caption of the figure (applies to all figure captions).

Thank you for this suggestion, however, we feel that highlighting the main result for each figure assists readers and clarifies our primary findings. Therefore, our preference is to retain this approach in the manuscript.

Lines 256-259: same comment (see above). As you always consider three levels of isolation, I suggest to uniform them in all the graphs (i.e., meadow, low isolation, high isolation), then you can specify in the caption to which distance they refer to.

Thank you for pointing out this inconsistency in the x-axis of Figure 1. We’ve updated all x-axes to be consistent in the graphs.

Lines 320-321: add references.

This sentence reflects the main finding of our study, as justified by the subsequent sentences in that initial paragraph of our discussion. 

Reviewer #2: The paper by Carroll and Freestone aim at evaluating how habitat isolation modifies community structure and mediates top-down and bottom-up control of primary producers in a marine seagrass system. The topic is interesting and the study is rightly conducted, as data are to be published, although some changes are needed to improve the paper. A list of comment/suggestions is reported below.

Line 95: “Our aim was to use seagrass meadows, a system of high conservation importance, to explore two questions”

Thank you, we’ve adopted this text.

Line 121: “temperature from 21 to 25.3°C, and salinity from 20 to 25 ppt” temporal reference for these values should be given

We’ve added a statement about the time period of the study.

Lines 124-142, Habitat isolation in natural seagrass: the sampling method should be better clarified.

How many replicates for each condition? Five? What is the logic to use transects if these latter are not considered in the analyses? Why do not consider three habitats and collect five replicates for each habitats? Transects should be justified or removed

Transects were used as a method of surveying the area for frequency and distribution of patches around the continuous meadow as well as determining appropriate distances for the artificial seagrass experiment. We’ve added a statement to the methods about the use of transects. We’ve reanalyzed our data to include transect as a random effect in our models. However, this factor was not significant (P>0.8) and was therefore removed from our final models. We now detail this in the manuscript.

Line 143, Habitat isolation and consumer-resource experiment: the use of artificial blades could potentially modify the recruitment pattern compared to natural blades. This aspect should be at least considered in the discussion.

We’ve added text regarding the use of ASUs in the discussion (now Lines 413-419 in the track changes document).

Lines 171: The analyses of nutrient levels in water column in ASU and control plots are needed to evaluate the nutrient increase; if the authors have no data, they should at least provide bibliographic references about the use of the same method to quantify the nutrient enhancing in seagrass systems (e.g Journal of Sea Research 63 (2010) 173-179; Scientific Reports (2017) 7, 13732)

We’ve incorporated the suggested Scientific Reports reference and another to the manuscript which is now Line 186.

6. PLOS authors have the option to publish the peer review history of their article (what does this mean?). If published, this will include your full peer review and any attached files.

Do you want your identity to be public for this peer review? For information about this choice, including consent withdrawal, please see our Privacy Policy.

Reviewer #1: No

Reviewer #2: No

---

## [Decision Letter · Decision Letter 1]

13 Jul 2023

Habitat isolation interacts with top-down and bottom-up processes in a seagrass ecosystem

PONE-D-23-00408R1

Dear Dr. Carroll,

We’re pleased to inform you that your manuscript has been judged scientifically suitable for publication and will be formally accepted for publication once it meets all outstanding technical requirements.

Kind regards,

Carlo Nike Bianchi

Academic Editor

PLOS ONE

Additional Editor Comments (optional):

Reviewers' comments:

Reviewer's Responses to Questions

**Comments to the Author**

1. If the authors have adequately addressed your comments raised in a previous round of review and you feel that this manuscript is now acceptable for publication, you may indicate that here to bypass the “Comments to the Author” section, enter your conflict of interest statement in the “Confidential to Editor” section, and submit your "Accept" recommendation.

Reviewer #1: All comments have been addressed

Reviewer #2: All comments have been addressed

2. Is the manuscript technically sound, and do the data support the conclusions?

Reviewer #1: Yes

Reviewer #2: Yes

3. Has the statistical analysis been performed appropriately and rigorously? 

Reviewer #1: Yes

Reviewer #2: Yes

4. Have the authors made all data underlying the findings in their manuscript fully available?

Reviewer #1: Yes

Reviewer #2: Yes

5. Is the manuscript presented in an intelligible fashion and written in standard English?

Reviewer #1: Yes

Reviewer #2: Yes

6. Review Comments to the Author

Reviewer #1: I have read and appreciated revisions the authors have made to the text but I still find some points of confusion that I suggest to clarify for the readers. It is now clear to me that transects were only used as a mean of identifying patches around the main body of the meadow and to define their distance from it: I believe that the supplementary figure 2 well explains the concept idea behind the study and it should be added to the main text. Eventually, patches could be defined in terms of size (what is the minimum size to define a patch or the average size order)

Talking about three field studies at the beginning of methods can be confusing, e.g. initially I thought the authors meant 3 replicates. I suggest specifying that these are three separate studies to investigate 1) Habitat isolation in natural seagrass; 2) Habitat isolation vs. consumer-resource 3) Recruitment.

Reviewer #2: (No Response)

7. PLOS authors have the option to publish the peer review history of their article (what does this mean?). If published, this will include your full peer review and any attached files.

Reviewer #1: No

Reviewer #2: No

---

## [Editor Report · Acceptance letter]

17 Jul 2023

PONE-D-23-00408R1 

Habitat isolation interacts with top-down and bottom-up processes in a seagrass ecosystem 

Dear Dr. Carroll:

I'm pleased to inform you that your manuscript has been deemed suitable for publication in PLOS ONE. Congratulations! Your manuscript is now with our production department. 

Kind regards, 

on behalf of

Dr. Carlo Nike Bianchi 

Academic Editor

PLOS ONE